# Effects of S-PBL in Maternity Nursing Clinical Practicum on Learning Attitude, Metacognition, and Critical Thinking in Nursing Students: A Quasi-Experimental Design

**DOI:** 10.3390/ijerph17217866

**Published:** 2020-10-27

**Authors:** Hae Kyoung Son

**Affiliations:** College of Nursing, Eulji University, Seongnam 13135, Korea; sonhk@eulji.ac.kr; Tel.: +82-31-740-7157; Fax: +82-31-740-7359

**Keywords:** nursing practicum, education, simulation, problem-based learning

## Abstract

Due to the coronavirus disease (COVID-19) pandemic, there are many restrictions in effect in clinical nursing practice. Since effective educational strategies are required to enhance nursing students’ competency in clinical practice, this study sought to evaluate the effectiveness of simulation problem-based learning (S-PBL). A quasi-experimental control group pretest-post-test design was used. Nursing students were allocated randomly to the control group (*n* = 31) and the experimental group (*n* = 47). Students in the control group participated in a traditional maternity clinical practicum for a week, while students in the experimental group participated S-PBL for a week. The students in the experimental group were trained in small groups using a childbirth patient simulator (Gaumard^®^ Noelle^®^ S554.100, Miami, USA) based on a standardized scenario related to obstetric care. The students’ learning attitude, metacognition, and critical thinking were then measured via a self-reported questionnaire. Compared with the control group, the pre-post difference in learning attitude and critical thinking increased significantly (*p* < 0.01) in the experimental group. S-PBL was found to be an effective strategy for improving nursing students’ learning transfer. Thus, S-PBL that reflects various clinical situations is recommended to improve the training in maternal health nursing.

## 1. Introduction

With the recent advances in medical technology and improvements in the living standards and education level of medical consumers, the demand for quality nursing services has increased [1]. Therefore, it is essential for nurses to acquire specialized nursing knowledge and skills, and have the integrated nursing ability to solve the various health problems of their patients [2,3]. Nursing educators should provide students with a variety of nursing opportunities through the clinical nursing practicum, so that they can adequately assess patients and provide effective nursing care in a dynamic environment [4,5].

Nursing education is composed of two complementary parts: theoretical training and practical training [6]. A nursing clinical practicum is an essential nursing curriculum that serves as a bridge between theoretical nursing science and clinical practice [3]. It provides nursing students with opportunities to develop therapeutic relationships with a variety of patients. In addition, they are equipped with the ability to solve real patient problems based on the theoretical knowledge learned in the classroom, thereby increasing nursing competency [3]. Since nursing clinical practice is on-the-job training that is required for nursing students to acquire basic competency as professional nurses, effective training strategies have been continuously discussed in the field of nursing education.

Due to the coronavirus disease (COVID-19) pandemic, contact between nursing students and patients as well as healthcare workers in the nursing field practice has become more restricted. The resulting passive form of the nursing clinical practicum has led to issues in the quality of nursing education. Thus, educational alternatives are needed. In view of the development of patient rights and safety protocols, as well as a lack of clinical educators, the nursing clinical practicum has also increased in student observation as opposed to direct nursing opportunities [7].

When nursing subjects deal with labor pains and childbirth, it is difficult to predict the nature of delivery, and emergency situations frequently arise. Thus, the opportunity for nursing students to provide direct nursing in such cases is limited, and the maternity nursing clinical practicum is more difficult than other nursing practices. Moreover, the number of deliveries in South Korea—which are the key cases in a maternity nursing clinical practicum—has sharply declined due to the very low birth rate. As it is difficult to achieve nursing learning goals using traditional teaching, rote memorization, and limited clinical practicum, there is an urgent need for alternatives to supplement the clinical nursing practicum. For students to acquire nursing competency as nursing professionals, a variety of educational methods that apply nursing theories and knowledge to actual situations must be applied.

Problem-based learning (PBL) is an alternative educational method that was first suggested in the medical field to overcome one of the problems of traditional medical education—the failure to develop the problem-solving ability required in clinical settings [8]. Simulation problem-based learning (S-PBL), an educational method that allows nursing students to repeatedly experience the process of solving nursing problems through interactions with a human patient simulator in an environment similar to clinical settings, is particularly useful for improving nursing competency [9].

Simulations using a high-fidelity maternal simulator that can reproduce patient characteristics in a laboratory similar to a clinical setting are effective in improving nursing students’ competency through more practical experiences [10,11]. As nursing simulations have recently been recognized as an effective way to supplement the limitations of a clinical nursing practicum, the number of studies regarding the effects of nursing education using simulations has increased [12]. However, the current nursing simulation practicum is used to apply and evaluate only partial nursing skills in consideration of the characteristics of a respective clinical practicum. Since S-PBL needs to be further developed and applied, this study aimed to provide nursing students with opportunities to provide nursing care during delivery, from phase 1 delivery to phase 4 delivery, using a high-fidelity maternal birth simulator.

An advantage of S-PBL is its ability to clearly identify and evaluate the learning transfer status of nursing students and provide them with feedback. In general, learning transfer is not limited to simply knowing, but also refers to putting knowledge into action. With nursing students, learning transfer refers to applying learned nursing knowledge to actual nursing situations. Kirkpatrick designed a conceptual framework for evaluating training programs. This training evaluation model provides an important systematic perspective regarding learning transfer, and is an influential tool that has been applied to many subsequent sub studies on the matter [13]. Kirkpatrick first published his model in 1959, updated it in 1975, and then again in 1993. This study intended to use Kirkpatrick’s evaluation model as the basis for its theoretical framework (Figure 1). The theoretical framework consists of four stepwise evaluations: (1) the reaction evaluation, or the evaluation of the nursing students’ motivation for S-PBL participation and understanding; (2) the learning evaluation, or the evaluation of principles, facts, and skills gained by learners as the learners’ advance preparation; (3) the behavior evaluation, or the evaluation wherein learners self-reflect and the professor evaluates whether the lessons learned are being transferred to actual performance and used in S-PBL; and (4) the result evaluation, or the evaluation of changes in learning attitude, metacognition, and critical thinking based on the results of the learners’ change in performance in S-PBL [14]. As demonstrated in Figure 1, the S-PBL employed in this study was designed to evaluate knowledge using a paper-based quiz and through “Questions & Answers” (Q&A) during the learning evaluation stage. Similarly, skill was evaluated using an S-PBL checklist during the behavior evaluation stage, wherein the instructor evaluated whether skills were being applied properly based on the learners’ self-reflection and acquired knowledge. At the result evaluation stage, the final results of the training were analyzed. Based on Kirkpatrick’s model, this study intended to evaluate learning attitudes, metacognition, and critical thinking in nursing students, and to confirm the learning transfer into S-PBL, and not simply ending with nursing knowledge. It is expected that short-term achievements will be linked to the nursing competency of professional nurses after graduation, which is a future achievement goal of nursing education for nursing students.

### 1.1. Aim(s)

This study aimed to investigate the effects of S-PBL, as part of a maternity nursing clinical practicum, on learning attitude, metacognition, and critical thinking in nursing students. More specifically, the objectives of this study were as follows:

First, determining the levels of learning attitude, metacognition, and critical thinking in nursing students;

Second, identifying the correlation between learning attitude, metacognition, and critical thinking in nursing students; and

Third, identifying the effects of learning attitude, metacognition, and critical thinking on learning transfer during the maternity nursing clinical practicum using S-PBL

### 1.2. Research Hypotheses

The research hypotheses that this study aimed to test were as follows:

First, the maternity nursing clinical practicum using S-PBL will lead to an improvement in learning attitude in nursing students;

Second, the maternity nursing clinical practicum using S-PBL will lead to an improvement in metacognition in nursing students; and

Third, the maternity nursing clinical practicum using S-PBL will lead to an improvement in critical thinking in nursing students.

## 2. Materials and Methods

### 2.1. Research Design

This study used a quasi-experimental design with a control group to compare the differences in learning attitude, metacognition, and critical thinking between nursing students undergoing the traditional maternity nursing clinical practicum and those undergoing S-PBL (Figure 2).

### 2.2. Subjects and Setting

The participants in this study were third-year students at a nursing college located in S City, South Korea, and were selected through convenience sampling from 98 nursing college students participating in the maternity nursing clinical practicum during the semester. The participants must have completed a maternity nursing course, which is a mandatory subject, and acquired basic skills related to maternity nursing, before participating in this study. Those with insufficient literacy skills were excluded from this study because data were collected using a survey in which the participants were supposed to read and answer the questionnaire.

The minimum required number of samples for this study was calculated based on a significance level of *α* = 0.05, power of 0.80, and effect size of 0.8 [15,16] using the G*Power 3.1 program. The results revealed that the minimum required number of samples for this study was 52 participants, with 26 in each group. The number of nursing students participating in the traditional maternity nursing clinical practicum or S-PBL varied between 12 and 15 on a weekly basis. A convenience sample of 78 nursing students were randomly allocated to the experimental group or control group considering the discrepancy between clinical practicum period and data collecting period (*n* = 20). The participants were collected considering the potential dropout rate during data collection. A total of 47 participants in the experimental group and 31 participants in the control group were included in the final analysis, meeting the minimum required number of samples (Figure 3).

### 2.3. Interventions

The research procedure was as follows: The participants were randomly assigned to either the control group or the experimental group, and participated in the traditional maternity nursing clinical practicum (1 week) or S-PBL (1 week), respectively.

In the traditional maternity clinical practicum, nursing students participated in maternity nursing clinical practice in the delivery rooms, obstetrics-gynecology ward, and operating rooms at a general hospital so that they can experience nursing care for pregnancy, delivery, and postpartum nursing, or surgical operations, such as cesarean section and myomectomy, for one week. Two to four nursing students per training department were assigned, and they participated in the traditional maternity clinical practicum by observing clinical nurses’ nursing activities, analyzing nursing cases, and reviewing the lessons learned during the clinical practice at conferences.

With the S-PBL, nursing students were provided with nursing knowledge in a simulation room at the university for one week, and performed simulation-based nursing activities in small groups. Three to four students were randomly assigned per group, after which they were instructed to analyze the clinical cases presented by the instructors, derive nursing problems, and directly apply the lessons they had learned in nursing care using a high-fidelity simulator (Gaumard^®^ Noelle^®^ S554.100, Miami, USA) that reproduced nursing situation. In particular, they were instructed to apply basic skills performance based on the nursing care case scenarios of phases 1, 2, 3, and 4 delivery. The basic skills performance included height of fundus measurement, abdominal circumference measurement, Leopold’s maneuvers, fetal monitoring, fetal heart sound auscultation via fetoscope and Doppler, nitrazine test, nursing care for uterine contraction and delivery pain, vertex delivery mechanism, Ritgen’s maneuver, fetal and placental expulsion, umbilical ligation, postpartum uterine contraction, and bleeding risk assessment. The S-PBL was developed by the researcher, based on the educational content that was delivered to students through a maternity nursing course intended for preview. While developing the S-PBL, consultation was sought from a nurse with extensive clinical nursing experience. Each group was instructed to perform the basic skills performance using the high-fidelity simulator based on clinical cases, with a running time of approximately 15–20 min. The instructor evaluated the learning transfer of the basic skills performance, which was learned through the operational use of maternity nursing simulations and the maternity nursing course using a structured evaluation checklist. During debriefing, after the group activities had been completed, the participants were also asked to complete an evaluation sheet consisting of a self-evaluation and self-reflection on the group activities. After those in the control and experimental groups completed the maternity nursing clinical practicum and the S-PBL, respectively, for one week, as well as the post-test, they were instructed to interchangeably participate in the S-PBL or the clinical nursing practicum regardless of this study.

### 2.4. Assessments

#### 2.4.1. Learning Attitude

In this study, learning attitude was measured using a scale modified by Hwang and Kim [17] from a Learning Attitude Measuring Scale developed by the Korea Educational Development Institute [18]. At the time of its development, the Learning Attitude Measuring Scale consisted of a total of 40 items, with 10 items regarding self-concept about concerned courses; 15 items regarding attitudes, such as interest, sense of purpose, and achievement motivation; and 15 items regarding learning habits, such as attention, self-learning, and exercising learning skills. The scale was then modified into a 16-item scale for nursing courses by Hwang and Kim [17]. The 16-item scale is a 5-point Likert scale ranging from 1 for ‘not at all’ to 5 for ‘always,’ with the total score ranging from a minimum of 5 points to a maximum of 80. A higher score indicates a better learning attitude in nursing students. The reliability of the original 40-item scale was Cronbach’s *α* = 0.83 [18], while the reliability of the scale by Hwang and Kim was Cronbach’s *α* = 0.84 [17], and its reliability in this study was Cronbach’s *α* = 0.82.

#### 2.4.2. Metacognition

Metacognition refers to nursing students’ awareness of their thinking processes in the learning process and their resulting ability to plan, check, and control the whole learning process [19]. Metacognition is a key element in the strategic aspect of the problem-solving process [20]. In this study, metacognition was measured using a tool that was originally developed by Klein [21] and later modified by Lee and Son [22]. This tool consists of three subdomains (cognitive strategy, planning, and self-checking), has 15 items, and is a 4-point Likert scale ranging from 1 for ‘not at all’ to 4 for ‘strongly agree.’ A higher score indicates a higher level of metacognition in nursing students. The reliability of the tool was Cronbach’s *α* = 0.91 in the study by Lee and Son [22], and its reliability in this study was Cronbach’s *α* = 0.86.

#### 2.4.3. Critical Thinking

Critical thinking refers to the personal disposition and habits that nursing students use to solve problems and make decisions. In this study, it was measured using a tool developed for nursing courses by Yoon [23]. In most domestic nursing studies, critical thinking was measured using the California Critical Thinking Disposition Inventory (CCTDI), or the tool developed by Yoon. However, the use of the CCTDI is restricted due to copyright issues, and the reliability in most previous studies that used the tool by Yoon was reported to be good [24]. The tool consists of a total of 27 items on a 5-point scale ranging from 1 for ‘not at all’ to 5 for ‘strongly agree.’ As the scores for two items (nos. 4 and 14) are inversely summed, the total score ranges from a minimum of 27 points to a maximum of 135. A higher score indicates a higher level of critical thinking in nursing students. The reliability of the tool in Yoon’s study was Cronbach’s *α* = 0.84 [23], and its reliability in the present study was Cronbach’s *α* = 0.87.

### 2.5. Data Collection

The nursing students participated in the traditional maternity nursing clinical practicum or S-PBL from 1 June to 17 July 2020, and the effects of the interventions on their learning attitude, metacognition, and critical thinking before and after their participation were measured using structured questionnaires. The researchers explained the contents of this study to the participants before the study began. After the nursing students who agreed to voluntarily participate in this study provided their written consent, they were made to answer the survey questionnaire. All of the participants were given a token of appreciation (school supplies worth KRW 1500).

### 2.6. Statistical Analysis

Data analysis was performed using IBM SPSS Statistics version 22.0 (IBM Co., Armonk, NY, USA). The general characteristics of the participants were analyzed using descriptive statistics. The reliability of the measurement tools used was analyzed using Cronbach’s alpha coefficient. The correlation between the variables was analyzed using Pearson’s correlation coefficient, and the effects of the intervention were analyzed using a t-test. The correlation between factors was analyzed by effect size for the correlation coefficient, and the statistical significance level was set at a *p*-value < 0.05.

### 2.7. Ethical Considerations

This study was approved by the Institutional Review Committee of the nursing college to which the researcher belongs (EUIRBN2020-010). All participants voluntarily decided to participate in this study, which was conducted in accordance with the Declaration of Helsinki. Prior to the commencement of the study, the participants were fully informed that their personal information would be kept confidential and would not be used for purposes other than this study, and that they could stop participating at any time during the study. They were also fully informed that there would be no disadvantages caused by their refusal to participate in the study, eliminating any unnecessary tension related to study participation. The survey answered by the participants took around 10 min, and there were no side effects caused by their participation other than minimal fatigue. After the participants in the control and experimental groups participated in the traditional maternity nursing clinical practicum and the S-PBL, respectively, for one week, and completed the post-test, they were given the opportunity to interchangeably participate in S-PBL or the clinical nursing practicum regardless of this study.

## 3. Results

### 3.1. General Characteristics

The general characteristics of the participants are shown in Table 1. The mean ages of the experimental and control groups were 22.06 years (1.97) and 22.10 years (3.13), respectively. There were 30 female students (63.8%) and 17 male students (36.2%) in the experimental group, and 28 female students (90.3%) and three male students (9.7%) in the control group. In terms of motivation for their choice of nursing major, the number of those who reported that they chose their major of their own will was the highest, with 30 (63.8%) and 20 (64.5%) in the experimental and control groups, respectively. In terms of major satisfaction, the number of those who reported that their major satisfaction was high was the highest in both groups, with 27 (57.4%) and 19 (61.3%) in the experimental and control groups, respectively. In terms of satisfaction with clinical nursing practicum, the number of those who reported that satisfaction with their clinical practice was high in both groups, with 23 (48.9%) and 19 (61.3%) in the experimental and control groups, respectively. In terms of academic achievement in the preceding semester, the number of those who reported that their academic achievement was moderate was the highest in both groups, with 21 (44.7%) and 16 (51.6%) in the experimental and control groups, respectively. Regarding academic stress, the number of those who reported that academic stress was high was the highest in both groups, with 25 (53.2%) and 19 (61.3%) in the experimental and control groups, respectively. The general characteristics of the participants were found to be homogeneous at pre-test between the two groups, except for gender (*t* = 3.93, *p* = 0.004) (*p* > 0.05). In addition, the variables in this study, such as learning attitude (*t* = 0.26, *p* = 0.799), metacognition (*t* = 0.13, *p* = 0.901), and critical thinking (*t* = 0.64, *p* = 0.522), were also found to be homogeneous between the two groups.

### 3.2. Correlation between Variables

The correlations between the variables are shown in Table 2. Learning attitude was correlated with metacognition (*r* = 0.669, *p* < 0.01) and critical thinking (*r* = 0.678, *p* < 0.01), and metacognition was correlated with critical thinking (*r* = 0.629, *p* < 0.01).

### 3.3. Effects of S-PBL

The effects of S-PBL compared with the traditional maternity clinical practicum are shown in Table 3. Learning attitude (*t* = −3.94, *p* < 0.001) and critical thinking values (*t* = −2.78, *p* = 0.008) in the experimental group were significantly higher post-training, as compared to pre-training. Meanwhile, there was no significant change in learning attitude, metacognition, and critical thinking in the control group. In particular, post-training critical thinking was lower in the control group than in the pre-test critical thinking (*t* = 0.26, *p* = 0.798).

## 4. Discussion

In order to verify the effects of the S-PBL, this study compared the learning attitudes, metacognition, and critical thinking of the nursing students that participated in the S-PBL and those that participated in the traditional maternity nursing clinical practicum. The results showed that nursing students who had participated in the S-PBL had significantly better learning attitudes and critical thinking post-training, as compared to pre-training.

In a study by Hwang and Kim [17], no significant difference was found in the learning attitude and critical thinking of nursing students that used PBL compared to traditional lecture-based learning. However, the results of this study showed that the practice-oriented S-PBL significantly improved learning attitudes and critical thinking in nursing students. During the clinical practice period, the scope of nursing students’ nursing experiences varied depending on the training department and patient cases. Considering the limitations of the clinical practicum, S-PBL with the use of a high-fidelity maternal simulator, which is designed to reproduce the characteristics of various patients in a simulation laboratory that is similar to clinical settings, can provide nursing students with more practical nursing cases and direct nursing experience; as such, it is more effective in improving nursing competency than the traditional clinical practice method [10,11]. Additionally, our study had considered the effect of the learning style such as the participants’ self-reflection and instructor performance skill checklist during the behavior evaluation stage of S-PBL. Further study should evaluate the effect of various learning styles using a high-fidelity simulator.

This study showed that S-PBL improved critical thinking in nursing students. The ultimate goal of improving critical thinking is to nurture nurses who are critical thinkers, with critical thinking skills and disposition [24]. According to the results of previous studies, critical thinking in nursing students was closely correlated with nursing competency [25], which is consistent with the goal of nursing education. In other words, critical thinking is an expected outcome of a nursing education program and an essential component of baccalaureate nursing education [2]. Since there is a need for the integrated education of critical thinking, problem-solving, leadership, and teamwork in nursing students with various learning style [26], various learning methods that reflect such needs, such as S-PBL, should be considered in nursing education.

However, in this study, no significant difference was found in metacognition post-training, as compared to pre-training. Metacognition refers to learners’ awareness of their thinking process as they learn so that they can plan, check, and control the entire learning process [19]. Metacognition is a key element in the strategic aspects of the problem-solving process [20], and is closely related to problem-solving ability [27,28]. It has been found that metacognition is learned and enhanced by learners in practical problem-solving situations [20]. Considering that the nursing students in this study were exposed to S-PBL for the first time, and participated in S-PBL for a short period, it is thought that nursing students’ long-term and repeated experiences of S-PBL in nursing curricula can improve metacognition in the whole learning process and the problem-solving process. In other words, nursing students monitor the process of solving nursing problems through S-PBL, which goes through the process of developing new metacognitive knowledge or modifying existing metacognitive knowledge in another S-PBL. Through this self-regulated learning, metacognition is acquired. Therefore, further studies are needed to develop and apply various learning strategies using S-PBL in nursing education.

Since S-PBL-related studies are scarce in Asian countries compared to Europe and North America [1], the study of this S-PBL approach in the Republic of Korea is of great significance to nursing education and studies. In particular, this study is meaningful for having demonstrated the use of S-PBL in improving learning transfer in nursing students through the provision of comprehensive simulation experiences of nursing, during stages 1 to 4 of childbirth. Considering the difficulties and risks involved for nursing students in gaining direct experience of working with patients during the COVID-19 pandemic, the nursing simulation used in this study may offer an opportunity to gain the necessary nursing skills while also ensuring the safety of both the patient and the student. A lot of care went into planning how students will be placed and for how long, and informed consent was obtained from them before they started clinical practice during the study [29,30]. Furthermore, based on the results of this study, it is necessary to apply S-PBL using systematic and integrated maternity nursing simulations, so that nursing students can learn a variety of maternity nursing cases.

## 5. Limitations

This study has several limitations. Nursing students’ self-assessment of learning attitude, metacognition, and critical thinking using the structured questionnaires poses limitations such as subjectivity. In addition, it is uncertain whether the difference between the intervention group and the control group comes from the learning styles such as learners’ self-reflection or instructor’s performance skill checklist or the use of a high-fidelity simulator. However, this study was based on the most widely used training evaluation model in the world. Considering the problem of the short-term program, this study finally suggested a long-term S-PBL program with a variety of nursing cases specially tailored to get nursing students to transfer learning in nursing curricula.

## 6. Impact Statement

This study suggested that nursing colleges incorporate S-PBL into the learning process so that nursing students can strengthen their learning attitude, metacognition, and critical thinking in their practical nursing education.

## 7. Conclusions

In maternity nursing, where the delivery process and outcome are difficult to predict and emergency situations frequently occur, the direct nursing care provided by nursing students is limited. In view of the fact that the birth rate in South Korea is very low, it is necessary to apply systematic and integrated maternity nursing simulations through which nursing students can learn various cases of maternity nursing as alternatives to supplement clinical practice. Under the circumstances where the development and application of S-PBL is required, the results of this study confirmed the effects of S-PBL-based nursing practicum on learning attitude, metacognition, and critical thinking in nursing students using a high-fidelity maternal birth simulator related to the main nursing care to be provided during delivery from phase 1 to phase 4 deliveries. Through its findings, this study provided evidence for promoting S-PBL. Finally, to promote nursing students’ learning transfer through S-PBL, it is suggested to determine various aspects of the knowledge, skills, and attitudes of nursing students in more objective ways, and to consider various learning methodologies in practical nursing education.

## Figures and Tables

**Figure 1 ijerph-17-07866-f001:**
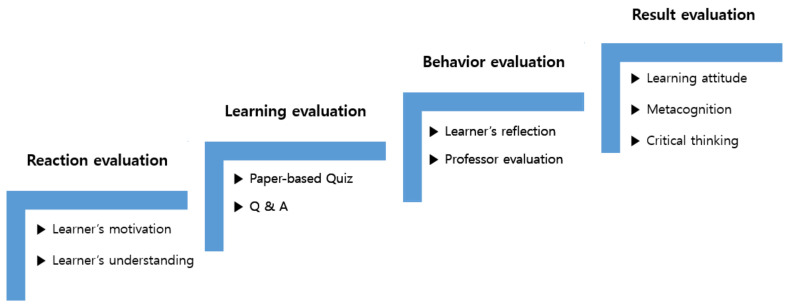
Framework of simulation problem-based learning (S-PBL).

**Figure 2 ijerph-17-07866-f002:**
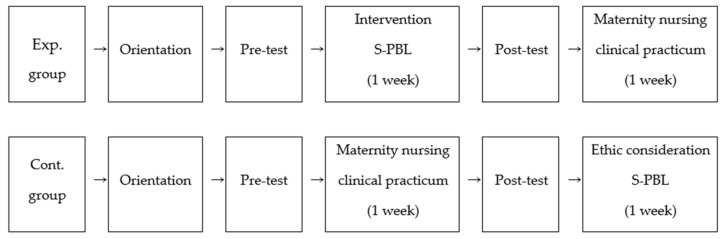
Research design. Exp. group = experimental group, Cont. group = control group; S-PBL = Simulation problem-based learning.

**Figure 3 ijerph-17-07866-f003:**
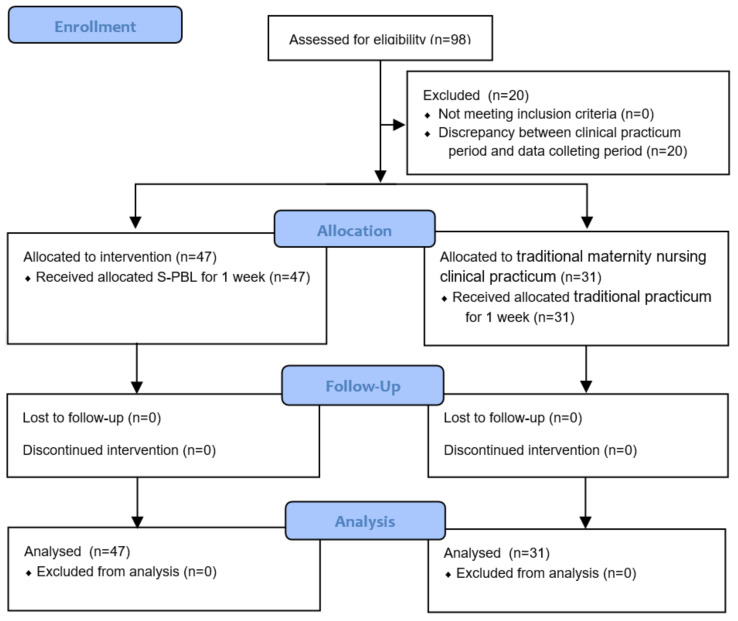
Research flow.

**Table 1 ijerph-17-07866-t001:** General Characteristics.

Characteristics	Categories/Range	Mean (SD)/Frequency (%)		t/*p*
		Exp. (*n* = 47)	Cont. (*n* = 31)	
Age (years)		22.06 (1.97)	22.10 (3.13)	0.05 (0.959)
Gender	female	30 (63.8)	28 (90.3)	2.98 (0.004)
	male	17 (36.2)	3 (9.7)	
Motivation for major choice	employment	9 (19.1)	5 (16.1)	0.20 (0.840)
	other’s recommendation	8 (17.0)	6 (19.4)	
	one’s own will	30 (63.8)	20 (64.5)	
Major satisfaction	very high	9 (19.1)	7 (22.6)	0.72 (0.476)
	high	27 (57.4)	19 (61.3)	
	moderate	11 (23.4)	5 (16.1)	
	low	0 (0.0)	0 (0.0)	
	very low	0 (0.0)	0 (0.0)	
Clinical practice satisfaction	very satisfied	14 (29.8)	4 (12.9)	1.51 (0.135)
	satisfied	23 (48.9)	19 (61.3)	
	moderate	10 (21.3)	7 (22.6)	
	unsatisfied	0 (0.0)	1 (3.2)	
	very unsatisfied	0 (0.0)	0 (0.0)	
Academic achievement	very high	3 (6.4)	5 (16.1)	0.32 (0.751)
	high	10 (21.3)	3 (9.7)	
	moderate	21 (44.7)	16 (51.6)	
	low	11 (23.4)	4 (12.9)	
	very low	2 (4.3)	3 (9.7)	
Academic stress	very high	8 (17.0)	1 (3.2)	1.43 (0.156)
	high	25 (53.2)	19 (61.3)	
	moderate	13 (27.7)	10 (32.3)	
	low	1 (2.1)	0 (0.0)	
	very low	0 (0.0)	1 (3.2)	
Learning attitude		60.87 (6.51)	61.26 (6.55)	0.26 (0.799)
Metacognition		46.89 (6.15)	47.06 (5.46)	0.13 (0.901)
Critical thinking		103.66 (8.71)	105.13 (11.42)	0.64 (0.522)

Exp. group = experimental group, Cont. group = control group.

**Table 2 ijerph-17-07866-t002:** Correlation between variables.

	r (*p*)		
	Learning Attitude	Metacognition	Critical Thinking
Learning Attitude	1	0.669 (0.000) **	0.678 (0.000) **
Metacognition		1	0.629 (0.000) **
Critical Thinking			1

** *p* < 0.01.

**Table 3 ijerph-17-07866-t003:** Effects of S-PBL.

Characteristics	Mean (SD)					
	Exp. (*n* = 47)			Cont. (*n* = 31)		
	Pretest	Posttest	t/*p*	Pretest	Posttest	t/*p*
Learning Attitude	60.87 (6.51)	63.36 (6.58)	−3.94 (<0.001) **	61.26 (6.55)	62.23 (6.04)	−1.01 (0.319)
Metacognition	46.89 (6.15)	48.15 (5.92)	−1.91 (0.063)	47.06 (5.46)	48.74 (5.40)	−1.52 (0.140)
Critical Thinking	103.66 (8.71)	106.30 (10.51)	−2.78 (0.008) *	105.13 (11.42)	104.65 (11.38)	0.26 (0.798)

**p* < 0.05, ** *p* < 0.01, * Exp. group = experimental group, Cont. group = control group.

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
