# Peer review of "Effects of S-PBL in Maternity Nursing Clinical Practicum on Learning Attitude, Metacognition, and Critical Thinking in Nursing Students: A Quasi-Experimental Design"

_ijerph, 2020, doi:10.3390/ijerph17217866_

Round 1
Reviewer 1 Report
This manuscript has a systematic organization, ideas are presented clearly, details given in different parts of the manuscript are appropriate and there are good uses of diagrams and tables. This study had a vigorous research design, and results are introduced systematically. The author(s) can consider the following suggestions.
It is mentioned in line 70 – 72 on page 2 that ‘As nursing simulations have recently been recognized as an effective way to supplement the limitations of a clinical nursing practicum, the number of studies regarding the effects of nursing education using simulations has increased’. Results of past research studies have indicate the effectiveness of simulation-based nursing education in different specialisms (e.g. Hur, Park, Shin, Lim, Kim, Kim, Choi, & Choi, 2013; Kim, Ko, & Lee, 2012; Shin, 2014) and in maternity nursing (Chung, Kim, & Park, 2011; Song, Y. A., & Son, Y. J., 2013). In addition, many of these previous studies used a similar control-experimental design as this study (e.g. Hur, Park, Shin, Lim, Kim, Kim, Choi, & Choi, 2013; Kim, Ko, & Lee, 2012; Chung, Kim, & Park, 2011). There is a need for the authors to highlight the significance of this study. Were there any innovative elements in the study described compared to previous studies?
Kirkpatrick’s theoretical framework was adopted in this study because its four-step model was highly suitable for this study. However, no justifications were offered for the adoption of this dated model, which was proposed in 1976. Has this model been repeatedly tested? Are there any other more recent models available?
Some photos such as screen captures or diagrams on the ‘Simulation problem-based learning’ software would substantially help readers to understand how it is implemented. It would be helpful to readers if the questionnaire or sample items (if there are copyright issues) can be appended to the manuscript.
There are some minor inconsistencies in the presentation of entries in the reference list. An example is the use of comma (‘,’) after the journal titles and before the publication years. The author(s) are advised to proofread the manuscript and rectify problems in presentation.
Chung, C-W., Kim, H-S., & Park, Y-S. (2011). Effects of high-fidelity simulation-based education on maternity nursing. Perspectives in Nursing Science, 8(2), 86–96.
Hur, H. K., Park, S., Shin, Y. H., Lim, Y. M., Kim, G., Kim, K. Y., Choi, H. O., & Choi, J. H. (2013). Development and applicability evaluation of an emergent care management simulation practicum for nursing students. The Journal of Korean Academic Society of Nursing Education, 19(2), 228–240.
Kim, H. Y., Ko, E., & Lee, E. S. (2012). Effects of simulation-based education on communication skill and clinical competence in maternity nursing practicum. Korean Journal of Women Health Nursing, 18(4), 312–320.
Shin, H. (2014). Evaluation of an integrated simulation courseware in a pediatric nursing practicum. Journal of Nursing Education, 53(10), 589–594.
Song, Y. A., & Son, Y. J. (2013). Effects of simulation-based practice education for core skill of maternity nursing. Korean Parent-Child Health Journal, 16(1), 37–44.
Author Response
Dear.
Thank you for your valuable feedback for the improvement of this article.
Please see the attachment.
If anything else is required with regard to this, I will be glad to consider the same.
Sincerely.

Reviewer 2 Report
This is a very interesting and novel study. Please find below some comment which may be useful for the improvement of the final version:
- Research hypotheses are clearly described. I suggest that you include the aim and specific objectives as well.
- Last step of Figure 2 should be clearly described. I am not sure why researchers included this step in the research process. If they think it is valuable then you need to explain.
- How intervention was developed (steps, framework etc) need to be explained.
- The aim of every intervention is to increase effectiveness, knowledge etc. Authors describe the effect of intervention on attitudes, metacognition and critical thinking. What about actual increase in knowledge and/ or skills which should be the main interest of this manuscript?
- In the discussion sections authors need to present other studies in other clinical setting that may have use same or similar methodology.
Author Response

(The authors gave the same response as above.)

Reviewer 3 Report
Thank you for the opportunity to review your paper. This is an interesting study using a quasi-experimental design. This study, which examined the effect of simulation problem-based learning, might contribute to improving the clinical training program. Under the COVID-19 pandemic, its significance has been increasing more than in the past. Generally, this paper was prepared well. However, I think this paper needs to revise because it has several major concerns following.
1.In the Materials and Methods, the sampling, and the allocation of the subjects were unclear. These points were described by the text. However, it was complicated. Therefore, in addition to the figure of the research design, using a flow chart would be recommended to explain these points.
2.In the interventions, I guessed there was a difference in not only using the simulator but the student's learning style between the intervention group and the control group. This study might evaluate the effect of the learning styles, not the use of the simulator. Did the students in the control group have been taken an evaluation of their performance using a structured evaluation checklist? And, were they asked to complete an evaluation sheet consisting of self-evaluation and self-reflection on their activities? If different learning styles between the two groups, you need to discuss this issue.
3.In the Discussion, the authors' comment: "The results showed, among others, that learning attitude and critical thinking were significantly increased in the nursing students that participated in the S-PBL compared to those that participated in the traditional maternity nursing clinical practicum (L273-276)" was not appropriate because the authors analyzed the differences in the three items between pre-training and post-training in each group but not did between the intervention group and the control group. The authors must directly compare the changes from pre-training to post-training between the two groups in each item.
4.In the Discussion, the authors' comment: "This provides evidence for improving nursing students’ competency (L276-277)" was possibly overinterpreted because of no direct comparison between the two groups mentioned above, and the short-term education program. The authors showed that the students' scores in the learning attitudes and critical thinking increased after the S-PBL training program. However, the one-week clinical training program might not enough to get the competency. In addition, the definition of the competency in this study was unclear. If the competency wants to be discussed, the authors need to make any comment concerning these issues.
5.In the Conclusions, the authors would be recommended to make a short comment concerning COVID-19. The authors mentioned it in the Abstract and the Introduction. It is recommended to ensure consistency.
Author Response

(The authors gave the same response as above.)

Round 2
Reviewer 3 Report
Thank you for replying carefully to the comments but the minor revisions were not enough. In addition, this paper did not have a study limitation section. You must describe a study limitation concerning the interpretation of the study findings.
Providing Figure 3 was not enough to reply to the comment about the sampling and the allocation. Which was the sample size in this study 98 subjects (at Line150) or 78 subjects (in Fig 3)?. Was it correct that there was no excluded sample? I think that this study has had the 20 subjects excluded from this study if N=98 was correct. How did you allocate these subjects to the two groups? Was it correct that there was no loss to follow-up? I think that this study has possibly any loss to follow-up because there was a difference in the number of subjects between the two groups. You need to respond to these questions. In addition, if there were any losses to follow-up, you must make any comment about the drop-out.
I could not find the text about the different learning styles in the Discussion section. You must make an additional comment about this issue as a study limitation.
Although you have changed from competency to transfer, I think the text: "This provides evidence of an improvement in nursing students’ learning transfer.", was too strong. It was unclear which this interpretation was supported by the evidence in this study or not because there were several limitations already mentioned. You must reconsider this sentence. In addition, you need to make an additional comment about the problem of the short-term program to get a competency or transfer as a study limitation.
Author Response

(The authors gave the same response as above.)
